# UniXGrad: A Universal, Adaptive Algorithm with Optimal Guarantees for Constrained Optimization

**Ali Kavis**[*]
EPFL
ali.kavis@epfl.ch

Kfir Y. Levy[*]
Technion
kfirylevy@technion.ac.il

Francis Bach
INRIA
francis.bach@inria.fr

Volkan Cevher
EPFL
volkan.cevher@epfl.ch

## Abstract

We propose a novel adaptive, accelerated algorithm for the stochastic constrained convex optimization setting. Our method, which is inspired by the Mirror-Prox method, *simultaneously* achieves the optimal rates for smooth/non-smooth problems with either deterministic/stochastic first-order oracles. This is done without any prior knowledge of the smoothness nor the noise properties of the problem. To the best of our knowledge, this is the first adaptive, unified algorithm that achieves the optimal rates in the constrained setting. We demonstrate the practical performance of our framework through extensive numerical experiments.

## 1 Introduction

Stochastic constrained optimization with first-order oracles (SCO) is critical in machine learning. Indeed, the scalability of classical machine learning tasks, such as support vector machines (SVMs), linear/logistic regression and Lasso, rely on efficient *stochastic* optimization methods. Importantly, generalization guarantees for such tasks often rely on constraining the set of possible solutions. The latter induces simple solutions in the form of low norm or low entropy, which in trun enables to establish generalization guarantees.

In the SCO setting, the optimal convergence rates for the cases of non-smooth and smooth objectives are given by $\mathcal{O}(GD/\sqrt{T})$ and $\mathcal{O}(LD^2/T^2 + \sigma D/\sqrt{T})$, respectively; where $T$ is the total number of (noisy) gradient queries, $L$ is the smoothness constant of the objective, $\sigma^2$ is the variance of the stochastic gradient estimates, $D$ is the effective diameter of the decision set, and $G$ is a bound on the magnitude of gradient estimates. These rates cannot be improved without additional assumptions.

The optimal rate for the non-smooth case may be obtained by the current state-of-the-art optimization algorithms, such as Stochastic Gradient Descent (SGD), AdaGrad [Duchi et al., 2011], Adam [Kingma and Ba, 2014], and AmsGrad [Reddi et al., 2018]. However, in order to obtain the optimal rate for the smooth case, one is required to use more involved *accelerated* methods such as [Hu et al., 2009, Lan, 2012, Xiao, 2010, Diakonikolas and Orecchia, 2017, Cohen et al., 2018, Deng et al., 2018].

Unfortunately, all of these accelerated methods require a-priori knowledge of the smoothness parameter $L$, as well as the variance of the gradients $\sigma^2$, creating a setup barrier for their use in practice. As a result, accelerated methods are not very popular in machine learning tasks.

This work develops a new *universal* method for SCO that obtains the optimal rates in both smooth and non-smooth cases, *without any prior knowledge regarding the smoothness of the problem L, nor*

---

[*]Equal contribution

*the noise magnitude* $\sigma$. Such universal methods that implicitly adapt to the properties of the learning objective may be very beneficial in practical large-scale problems where these properties are usually unknown. To our knowledge, this is the first work that achieves this desiderata in the constrained SCO setting.

**Our contributions in the context of related work**   For the unconstrained setting, Levy et al. [2018] and Cutkosky [2019] have recently presented a universal scheme that obtains (almost) optimal rates for both smooth and non-smooth cases.

More specifically, Levy et al. [2018] designs AcceleGrad—a method that obtains respective rates of $\mathcal{O}\left(GD\sqrt{\log T}/\sqrt{T}\right)$ and $\mathcal{O}\left(L\log LD^2/T + \sigma D\sqrt{\log T}/\sqrt{T}\right)$. Unfortunately, this result only holds for the unconstrained setting, and the authors leave the *constrained* case as an open problem.

An important progress towards this open problem is achieved only recently by Cutkosky [2019], who proves suboptimal respective rates of $\mathcal{O}\left(1/\sqrt{T}\right)$ and $\mathcal{O}\left(D^2L/T^{3/2} + \sigma D/\sqrt{T}\right)$ for SCO in the constrained setting.

Our work completely resolves the open problem in Levy et al. [2018], Cutkosky [2019], and proposes the first *universal* method that obtains respective *optimal* rates of $\mathcal{O}\left(GD/\sqrt{T}\right)$ and $\mathcal{O}\left(D^2L/T^2 + \sigma D/\sqrt{T}\right)$ for the constrained setting. When applied to the unconstrained setting, our analysis tightens the rate characterizations by removing the unnecessary logarithmic factors appearing in [Levy et al., 2018, Cutkosky, 2019].

Our method is inspired by the Mirror-Prox method [Nemirovski, 2004, Rakhlin and Sridharan, 2013, Diakonikolas and Orecchia, 2017, Bach and Levy, 2019], and builds on top of it using additional techniques from the online learning literature. Among, is an adaptive learning rate rule [Duchi et al., 2011, Rakhlin and Sridharan, 2013], as well as recent online-to-batch conversion techniques [Levy, 2017, Cutkosky, 2019].

The paper is organized as follows. In the next section, we specify the problem setup, and give the necessary definitions and background information. In Section 3, we motivate our framework and explain the general mechanism. We also introduce the convergence theorems with proof sketches to highlight the technical novelties. We share numerical results in comparison with other adaptive methods and baselines for different machine learning tasks in Section 4, followed up with conclusions.

## 2   Setting and preliminaries

**Preliminaries.**   Let $\|\cdot\|$ be a general norm and $\|\cdot\|_*$ be its dual norm. A function $f : \mathcal{K} \mapsto \mathbb{R}$ is $\mu$-*strongly convex* over a convex set $\mathcal{K}$, if for any $x \in \mathcal{K}$ and any $\nabla f(x)$, a subgradient of $f$ at $x$,

$$f(x) - f(y) - \langle \nabla f(y), x - y \rangle \geq \frac{\mu}{2}\|x - y\|^2, \quad \forall x, y \in \mathcal{K} \tag{1}$$

A function $f : \mathcal{K} \mapsto \mathbb{R}$ is $L$-*smooth* over $\mathcal{K}$ if it has $L$-Lipschitz continuous gradient, i.e.,

$$\|\nabla f(x) - \nabla f(y)\|_* \leq L\|x - y\|, \quad \forall x, y \in \mathcal{K}. \tag{2}$$

Consider a 1-strongly convex differentiable function $\mathcal{R} : \mathcal{K} \to \mathbb{R}$. The Bregman divergence with respect to a distance-generating function $\mathcal{R}$ is defined as follows $\forall x, y \in \mathcal{K}$,

$$D_{\mathcal{R}}(x, y) = \mathcal{R}(x) - \mathcal{R}(y) - \langle \nabla \mathcal{R}(y), x - y \rangle \ . \tag{3}$$

An important property of Bregman divergence is that $D_{\mathcal{R}}(x, y) \geq \frac{1}{2}\|x - y\|^2$ for all $x, y \in \mathcal{K}$, due to the strong convexity of $\mathcal{R}$.

**Setting**   This paper focuses on (approximately) solving the following constrained problem,

$$\min_{x \in \mathcal{K}} f(x) \ , \tag{4}$$

where $f : \mathcal{K} \mapsto \mathbb{R}$ is a convex function, and $\mathcal{K} \subset \mathbb{R}^d$ is a compact convex set.

We assume the availability of a first order oracle for $f(\cdot)$, and consider two settings: a deterministic setting where we may access exact gradients, and a stochastic setting where we may only access unbiased (noisy) gradient estimates. Concretely, we assume that by querying this oracle with a point $x \in \mathcal{K}$, we receive $\tilde{\nabla} f(x) \in \mathbb{R}^d$ such,

$$\mathbb{E}\left[\tilde{\nabla} f(x) \big| x\right] = \nabla f(x) . \tag{5}$$

Throughout this paper we also assume the norm of the (sub)-gradient estimates is bounded by $G$, i.e,

$$\|\tilde{\nabla} f(x)\|_* \leq G, \qquad \forall x \in \mathcal{K} .$$

## 3 The algorithm

In this section, we present and analyze our **Uni**versal e**X**tra **Grad**ient (UniXGrad) method. We first discuss the Mirror-Prox (MP) algorithm of [Nemirovski, 2004], and the related Optimistic Mirror Descent (OMD) algorithm of [Rakhlin and Sridharan, 2013]. Later we present our algorithm which builds on top of the Optimistic Mirror Descent (OMD) scheme. Then in Sections 3.1 and 3.2, we present and analyze the guarantees of our method in nonsmooth and smooth settings, respectively.

Our goal is to optimize a convex function $f$ over a compact domain $\mathcal{K}$, and Algorithm 1 offers a framework for solving this template, which is inspired by the Mirror-Prox (MP) algorithm of [Nemirovski, 2004] and the Optimistic Mirror Descent (OMD) algorithm of [Rakhlin and Sridharan, 2013]. Let us motivate this particular template. Basically, the algorithm takes a step from $y_{t-1}$ to $x_t$, using first order information based on $y_{t-1}$. Then, it goes back to $y_{t-1}$ and takes another step, but this time, gradient information relies on $x_t$. Each step is a generalized projection with respect to Bregman divergence $D_{\mathcal{R}}(\cdot, \cdot)$.

---

**Algorithm 1** Mirror-Prox Template

---

**Input:** Number of iterations $T$, $y_0 \in \mathcal{K}$, learning rate $\{\eta_t\}_{t \in [T]}$
1: **for** $t = 1, ..., T$ **do**
2: $\qquad x_t = \arg\min_{x \in \mathcal{K}} \langle x, M_t \rangle + \frac{1}{\eta_t} D_{\mathcal{R}}(x, y_{t-1})$
3: $\qquad y_t = \arg\min_{y \in \mathcal{K}} \langle y, g_t \rangle + \frac{1}{\eta_t} D_{\mathcal{R}}(y, y_{t-1})$
4: **end for**

---

Now, let us explain the salient differences between UniXGrad and MP as well as OMD using the particular choices of $M_t$, $g_t$ and the distance-generating function $\mathcal{R}$.

Optimistic Mirror Descent takes $g_t = \nabla f(x_t)$ and computes $M_t = \nabla f(x_{t-1})$, i.e., based on gradient information from previous iterates. This vector is available at the beginning of each iteration and the "optimism" arises in the case where $M_t \approx g_t$. When $M_t = \nabla f(y_{t-1})$ and $g_t = \nabla f(x_t)$, the template is known as the famous Mirror-Prox algorithm. One special case of Mirror-Prox is Extra-Gradient scheme [Korpelevich, 1976] where the projections are with respect to Euclidean norm, i.e. $\mathcal{R}(x) = 1/2 \|x\|_2^2$, instead of general Bregman divergences.

MP has been well-studied, especially in the context of variational inequalities and convex-concave saddle point problems. It achieves fast convergence rate of $\mathcal{O}(1/T)$ for this class of problems, however, in the context of smooth convex optimization, this is the standard slow rate [Nesterov, 2003]. To date, MP is not known to enjoy the accelerated rate of $\mathcal{O}(1/T^2)$ for smooth convex minimization.

We propose three modifications to this template, which are the precise choice of $g_t$ and $M_t$, the adaptive learning rate and the gradient weighting scheme.

**The notion of averaging:** In different interpretations of acceleration [Nesterov, 1983, Tseng, 2008, Allen Zhu and Orecchia, 2014], the notion of averaging is always central and we incorporate this notion via gradients taken at weighted average of iterates. Let us define the weight $\alpha_t = t$ and the following quantities

$$\bar{x}_t = \frac{\alpha_t x_t + \sum_{i=1}^{t-1} \alpha_i x_i}{\sum_{i=1}^t \alpha_i}, \qquad \tilde{z}_t = \frac{\alpha_t y_{t-1} + \sum_{i=1}^{t-1} \alpha_i x_i}{\sum_{i=1}^t \alpha_i}. \tag{6}$$

Then, UniXGrad algorithm takes $g_t = \nabla f(\bar{x}_t)$ and $M_t = \nabla f(\tilde{z}_t)$, which provides a naive interpretation of averaging. Our choice of $g_t$ and $M_t$ coincide with that of the accelerated Extra-Gradient scheme of Diakonikolas and Orecchia [2017]. While their decision relies on implicit Euler discretization of an accelerated dynamics, we arrive at the same conclusion as a direct consequence of our convergence analysis.

**Adaptive learning rate:** A key ingredients of our algorithm is the choice of adaptive learning rate $\eta_t$. In light of Rakhlin and Sridharan [2013], we define our lag-one-behind learning rate as

$$\eta_t = \frac{2D}{\sqrt{1 + \sum\limits_{i=1}^{t-1} \alpha_i^2 \|g_i - M_i\|_*^2}}, \tag{7}$$

where $D^2 = \sup_{x,y \in \mathcal{K}} D_{\mathcal{R}}(x, y)$ is the diameter of the compact set $\mathcal{K}$ with respect to Bregman divergences. Algorithm 2 summarizes our framework.

**Gradient weighting scheme:** We introduce the weights $\alpha_t$ in the sequence updates. One can interpret this as separating step size into learning rate and the scaling factors. It is necessary that $\alpha_t = \Theta(t)$ in order to achieve optimal rates, in fact we precisely choose $\alpha_t = t$. Also notice that they appear in the learning rate, compatible with the update rule.

---

**Algorithm 2** UniXGrad

---

**Input:** # of iterations $T$ , $y_0 \in K$, diameter $D$, weight $\alpha_t = t$, learning rate $\{\eta_t\}_{t \in [T]}$
1: **for** $t = 1, ..., T$ **do**
2: $\quad\quad x_t = \arg\min\limits_{x \in \mathcal{K}} \alpha_t \langle x, M_t \rangle + \frac{1}{\eta_t} D_{\mathcal{R}}(x, y_{t-1}) \quad\quad (M_t = \nabla f(\tilde{z}_t))$
3: $\quad\quad y_t = \arg\min\limits_{y \in \mathcal{K}} \alpha_t \langle y, g_t \rangle + \frac{1}{\eta_t} D_{\mathcal{R}}(y, y_{t-1}) \quad\quad (g_t = \nabla f(\bar{x}_t))$
4: **end for**
5: **return** $\bar{x}_T$

---

In the remainder of this section, we will present our convergence theorems and provide proof sketches to emphasize the fundamental aspects and novelties. With the purpose of simplifying the analysis, we borrow classical tools in the online learning literature and perform the convergence analysis in the sense of bounding "weighted regret". Then, we use a simple yet essential conversion strategy which enables us to *directly* translate our weighted regret bounds to convergence rates. Before we proceed, we will present the conversion scheme from weighted regret to convergence rate, by deferring the proof to Appendix. In a concurrent work, [Cutkosky, 2019] proves a similar online-to-offline conversion bound.

**Lemma 1.** *Consider weighted average $\bar{x}_t$ as in Eq. (6). Let $R_T(x_*) = \sum_{t=1}^{T} \alpha_t \langle x_t - x_*, g_t \rangle$ denote the weighted regret after T iterations, $\alpha_t = t$ and $g_t = \nabla f(\bar{x}_t)$. Then,*

$$f(\bar{x}_T) - f(x_*) \leq \frac{2R_T(x_*)}{T^2}.$$

### 3.1 Non-smooth setting

**Deterministic setting:** First, we will focus on the convergence analysis in the case of non-smooth objective functions with deterministic/stochastic first-order oracles. We will follow the regret analysis as in Rakhlin and Sridharan [2013] with essential adjustments that suit our weighted scheme and particular choice of adaptive learning rate.

**Remark 1.** *It is important to point out that we do not completely exploit the precise definitions of $g_t$ and $M_t$ in the presence of non-smooth objectives. As far as the regret analysis is concerned, it suffices that these quantities are functions of $\nabla f(\cdot)$ and that, as a corollary, their dual norm is upper bounded. However, in order to bridge the gap between weighted regret and the objective sub-optimality, i.e. $f(\bar{x}_T) - f(x_*)$, we require $g_t = \nabla f(\bar{x}_t)$.*

Now, we can exhibit our convergence bounds for the case of deterministic oracles.

**Theorem 1.** *Consider the constrained optimization setting in Problem ([4](#)), where $f : \mathcal{K} \to \mathbb{R}$ is a proper, convex and $G$-Lipschitz function defined over compact, convex set $\mathcal{K}$. Let $x^* \in \min_{x \in \mathcal{K}} f(x)$. Then, Algorithm [2](#) guarantees*

$$f(\bar{x}_T) - \min_{x \in \mathcal{K}} f(x) \leq \frac{7D\sqrt{1 + \sum_{t=1}^{T} \alpha_t^2 \left\| g_t - M_t \right\|_*^2} - D}{T^2} \leq \frac{6D}{T^2} + \frac{14GD}{\sqrt{T}}. \tag{8}$$

We establish the basis of our analysis through Lemma 1 and Corollary 2 of Rakhlin and Sridharan [2013]. Then, we build upon this base by exploiting the structure of the adaptive learning rate, the weights $\alpha_t$ and the bound on gradient norms to give adaptive convergence bounds.

**Stochastic setting:** Now, we further consider the case of stochastic gradients. We assume that the first-order oracles are unbiased (see Eq. ([5](#))). We want to emphasize that our stochastic setting is *not* restricted to the notion of additive noise, i.e. gradients corrupted with zero-mean noise. It essentially includes any estimate that recovers the full gradient in expectation, e.g. estimating gradient using mini batches. Additionally, we propagate the bounded gradient norm assumption to the stochastic oracles, such that $\|\tilde{\nabla} f(x)\|_* \leq G, \forall x \in \mathcal{K}$.

**Theorem 2.** *Consider the optimization setting in Problem ([4](#)), where $f$ is non-smooth, convex and $G$-Lipschitz. Let $\{x_t\}_{t=1,..,T}$ be a sequence generated by Algorithm [2](#) such that $g_t = \tilde{\nabla} f(\bar{x}_t)$ and $M_t = \tilde{\nabla} f(\tilde{z}_t)$. With $\alpha_t = t$ and learning rate as in Eq. ([7](#)), it holds that*

$$\mathbb{E}\left[f(\bar{x}_T)\right] - \min_{x \in \mathcal{K}} f(x) \leq \frac{6D}{T^2} + \frac{14GD}{\sqrt{T}}.$$

The analysis in the stochastic setting is similar to deterministic setting. The difference is up to replacing $g_t \leftrightarrow \tilde{g}_t$ and $M_t \leftrightarrow \tilde{M}_t$. With the bound on stochastic gradients, the same rate is achieved.

## 3.2 Smooth setting

**Deterministic setting:** In terms of theoretical contributions and novelty, the case of $L$-smooth objective is of greater interest. We will first start with the deterministic oracle scheme and then introduce the convergence theorem for the noisy setting.

**Theorem 3.** *Consider the constrained optimization setting in Problem ([4](#)), where $f : \mathcal{K} \to \mathbb{R}$ is a proper, convex and $L$-smooth function defined over compact, convex set $\mathcal{K}$. Let $x^* \in \min_{x \in \mathcal{K}} f(x)$. Then, Algorithm [2](#) ensures the following*

$$f(\bar{x}_T) - \min_{x \in \mathcal{K}} f(x) \leq \frac{20\sqrt{7}D^2 L}{T^2}. \tag{9}$$

**Remark 2.** *In the non-smooth setting, we assume that gradients have bounded norms. Our algorithm does **not** need to know this information, but it is necessary for the analysis in that case. However, when the function is smooth, neither the algorithm nor the analysis requires bounded gradients.*

*Proof Sketch (Theorem [3](#)).* We follow the proof of Theorem [1](#) until the point where we obtain

$$\sum_{t=1}^{T} \alpha_t \langle x_t - x_*, g_t \rangle \leq \frac{1}{2} \sum_{t=1}^{T} \eta_{t+1} \alpha_t^2 \left\| g_t - M_t \right\|_*^2 - \frac{1}{2} \sum_{t=1}^{T} \frac{1}{\eta_{t+1}} \left\| x_t - y_{t-1} \right\|^2 + D^2 \left( \frac{3}{\eta_{T+1}} + \frac{1}{\eta_1} \right).$$

By smoothness of the objective function, we have $\| g_t - M_t \|_* \leq L \| \bar{x}_t - \tilde{z}_t \|$, which implies $-\frac{1}{\eta_{t+1}} \| x_t - y_{t-1} \|^2 \leq -\frac{\alpha_t^2}{4L^2 \eta_{t+1}} \| g_t - M_t \|_*^2$. Hence,

$$\leq \frac{1}{2} \sum_{t=1}^{T} \left( \eta_{t+1} - \frac{1}{4L^2 \eta_{t+1}} \right) \alpha_t^2 \left\| g_t - M_t \right\|_*^2 + D^2 \left( \frac{3}{\eta_{T+1}} + \frac{1}{\eta_1} \right).$$

Now we will introduce a time variable to *characterize* the growth of the learning rate. Define $\tau^* = \max \left\{ t \in \{1, ..., T\} : \frac{1}{\eta_{t+1}^2} \leq 7L^2 \right\}$ such that $\forall t > \tau^*, \eta_{t+1} - \frac{1}{4L^2 \eta_{t+1}} \leq -\frac{3}{4} \eta_{t+1}$. Then,

$$\leq D \underbrace{\sum_{t=1}^{\tau^*} \frac{\alpha_t^2 \left\| g_t - M_t \right\|_*^2}{\sqrt{1 + \sum_{i=1}^t \alpha_i^2 \left\| g_i - M_i \right\|_*^2}} + \frac{D}{2}}_{(A)}$$

$$+ \underbrace{\frac{3D}{2} \left( \sqrt{1 + \sum_{t=1}^T \alpha_t^2 \left\| g_t - M_t \right\|_*^2} - \sum_{t=\tau^*+1}^T \frac{\alpha_t^2 \left\| g_t - M_t \right\|_*^2}{\sqrt{1 + \sum_{i=1}^t \alpha_i^2 \left\| g_i - M_i \right\|_*^2}} \right)}_{(B)},$$

where we wrote $\eta_{t+1}$ in open form and used the definition of $\tau^*$. To complete the proof, we will need the following lemma.

**Lemma 2.** *Let $\{a_i\}_{i=1,\dots,n}$ be a sequence of non negative numbers. Then, it holds that*

$$\sqrt{\sum_{i=1}^n a_i} \leq \sum_{i=1}^n \frac{a_i}{\sqrt{\sum_{j=1}^i a_j}} \leq 2\sqrt{\sum_{i=1}^n a_i}.$$

Please refer to [McMahan and Streeter, 2010, Levy et al., 2018] for the proof. We jointly use Lemma 2 and the bound on $\eta_{\tau^*+1}$ to upper bound terms (A) and (B) with $4\sqrt{7}D^2L$ and $6\sqrt{7}D^2L$, respectively. Lemma 1 immediately establishes the convergence bound. $\qquad\square$

**Stochastic setting:** Next, we will present our results for the stochastic extension. In addition to unbiasedness and boundedness, we will introduce another classical assumption: bounded variance,

$$E[\|\nabla f(x) - \tilde{\nabla} f(x)\|_*^2 | x] \leq \sigma^2, \quad \forall x \in \mathcal{K}. \tag{10}$$

The analysis proceeds along similar lines as its deterministic counterpart. However, we execute the analysis using auxiliary terms and attain the optimal accelerated rate without the log factors.

**Theorem 4.** *Consider the optimization setting in Problem* (4)*, where $f$ is L-smooth and convex. Let $\{x_t\}_{t=1,\dots,T}$ be a sequence generated by Algorithm 2 such that $g_t = \tilde{\nabla} f(\bar{x}_t)$ and $M_t = \tilde{\nabla} f(\tilde{z}_t)$. With $\alpha_t = t$ and learning rate as in* (7)*, it holds that*

$$\mathbb{E}\left[f(\bar{x}_T)\right] - \min_{x \in \mathcal{K}} f(x) \leq \frac{224\sqrt{14}D^2L}{T^2} + \frac{14\sqrt{2}\sigma D}{\sqrt{T}}.$$

*Proof Sketch (Theorem 4).* We start in the same spirit as the stochastic, non-smooth setting,

$$\sum_{t=1}^T \alpha_t \left\langle x_t - x_*, g_t \right\rangle \leq \underbrace{\sum_{t=1}^T \alpha_t \left\langle x_t - x^*, \tilde{g}_t \right\rangle}_{(A)} + \underbrace{\sum_{t=1}^T \alpha_t \left\langle x_t - x^*, g_t - \tilde{g}_t \right\rangle}_{(B)}.$$

Recall that term (B) is zero in expectation given $\bar{x}_t$. Then, we follow the proof steps of Theorem 1,

$$\sum_{t=1}^T \alpha_t \left\langle x_t - x_*, g_t \right\rangle \leq \frac{7D}{2} \sqrt{1 + \sum_{t=1}^T \alpha_t^2 \|\tilde{g}_t - \tilde{M}_t\|_*^2} - \frac{1}{2} \sum_{t=1}^T \frac{1}{\eta_{t+1}} \|x_t - y_{t-1}\|^2. \tag{11}$$

We will obtain $\|g_t - M_t\|_*^2$ from $\|x_t - y_{t-1}\|^2$ due to smoothness and the challenge is to handle $\|\tilde{g}_t - \tilde{M}_t\|_*^2$ and $\|g_t - M_t\|_*^2$ together. So let's denote, $B_t^2 := \min\{\|g_t - M_t\|_*^2, \|\tilde{g}_t - \tilde{M}_t\|_*^2\}$. Using this definition, we could declare an auxiliary learning rate which we will *only* use for the analysis,

$$\tilde{\eta}_t = \frac{2D}{\sqrt{1 + \sum\limits_{i=1}^{t-1} \alpha_i^2 B_i^2}}. \tag{12}$$

Clearly, for any $t \in [T]$ we have $-\frac{1}{\eta_{t+1}} \|g_t - M_t\|_*^2 \le -\frac{1}{\tilde{\eta}_{t+1}} B_t^2$. Also, we can write,

$$\|\tilde{g}_t - \tilde{M}_t\|_*^2 \le 2\|g_t - M_t\|_*^2 + 2\|\xi_t\|_*^2, \tag{13}$$

and,

$$\|\tilde{g}_t - \tilde{M}_t\|_*^2 \le 2B_t^2 + 2\|\xi_t\|_*^2.$$

Therefore, we could rewrite Eq. (11) as,

$$\sum_{t=1}^{T} \alpha_t \langle x_t - x_*, g_t \rangle \le \underbrace{\frac{7}{2} \sum_{t=1}^{T} \left( \tilde{\eta}_{t+1} - \frac{1}{28L^2 \tilde{\eta}_{t+1}} \right) \alpha_t^2 B_t^2 + \frac{7D}{2}}_{(A)} + \underbrace{\frac{7D}{\sqrt{2}} \sqrt{\sum_{t=1}^{T} \alpha_t^2 \|\xi_t\|_*^2}}_{(B)}.$$

Using Lemma 2 and defining a time variable $\tau_*$ in the sense of Theorem 3 (with correct constants), term (A) is upper bounded by $112\sqrt{14}D^2L$. By taking expectation conditioned on $\bar{x}_t$ and using Jensen's inequality, we could upper bound term (B) as $14\sigma DT^{3/2}/\sqrt{2}$, which leads us to the optimal rate of $224\sqrt{14}D^2L/T^2 + 14\sqrt{2}\sigma D/\sqrt{T}$ through Lemma 1. $\qquad\square$

# 4 Experiments

We compare performance of our algorithm for two different tasks against adaptive methods of various characteristics, such as AdaGrad, AMSGrad and AcceleGrad, along with a recent non-adaptive method AXGD. We consider a synthetic setting where we analyze the convergence behavior, as well as a SVM classification task on some LIBSVM dataset. In all the setups, we tuned the hyper-parameters of each algorithm by grid search. In order to compare the adaptive methods on equal grounds, AdaGrad is implemented with a scalar step size based on the template given by Levy [2017]. We implement AMSGrad exactly as it is described by Reddi et al. [2018].

## 4.1 Convergence behavior

We take the least squares problem with $L_2$-norm ball constraint for this setting, i.e., $\min_{\|x\|_2 < r} \frac{1}{2n} \|Ax - b\|_2^2$, where $A \in \mathbb{R}^{n \times d}$, $A \sim \mathcal{N}(0, \sigma^2 I)$ and $b = Ax^\natural + \epsilon$ such that $\epsilon$ is a random vector $\sim \mathcal{N}(0, 10^{-3})$. We pick $n = 500$ and $d = 100$. For the rest of this section, we refer to the solution of *constrained* problem as $x_*$.

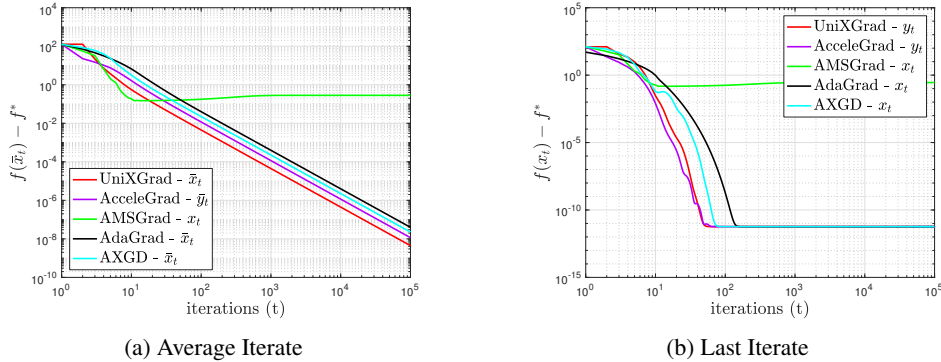

(a) Average Iterate
(b) Last Iterate

Figure 1: Convergence rates in the **deterministic** oracle setting when $x_* \in \text{Boundary}(\mathcal{K})$

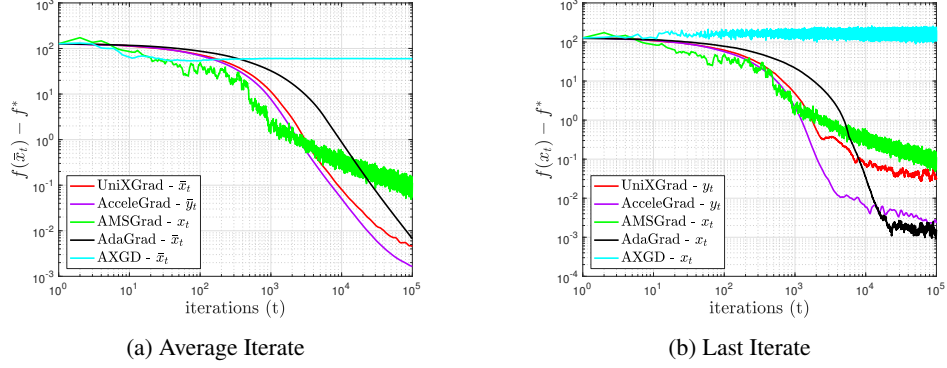

(a) Average Iterate                          (b) Last Iterate

Figure 2: Convergence rates in the **stochastic** oracle setting when $x_* \in \text{Boundary}(\mathcal{K})$

In Figure 1 and 2, we present the convergence rates under deterministic and stochastic oracles, and we pick a problem in which the solution is on the boundary of the constraint set, i.e., $x_* \in \text{Boundary}(\mathcal{K})$. In this setting, our algorithm shows matching performance in comparison with other methods. AXGD has convergence issues in the stochastic setting, as it only handles additive noise and their step size routine does not seem to be compatible with stochastic gradients. Another key observation is that AMSGrad suffers a decrease in its performance when the solution is on the boundary of the set.

## 4.2 SVM classification

In this section, we will tackle SVM classification problem on "breast-cancer" data set taken from LIBSVM. We try to minimize squared Hinge loss with $L_2$ norm regularization. We split the data set as training and test sets with 80/20 ratio. The models are trained using random mini batches of size 5. Figure 3 demonstrates convergence rates and test accuracies of the methods. They represent the average performance of 5 runs, with random initializations. For UniXGrad, AcceleGrad and AXGD, we consider the performance with respect to the average iterate $\bar{x}_t$ as it shows a more stable behavior, whereas AdaGrad and AMSGrad are evaluated based on their last iterates. AXGD, which has poor convergence behavior in stochastic setting due to its step size rule, shows the worst performance both in terms of convergence and generalization. UniXGrad, AcceleGrad, AdaGrad and AMSGrad achieve comparable generalization performances to each other. AMSGrad achieves a slightly better performance as it has diagonal preconditioner which translates to per-coordinate learning rate. It could possibly adapt to the geometry of the optimization landscape better.

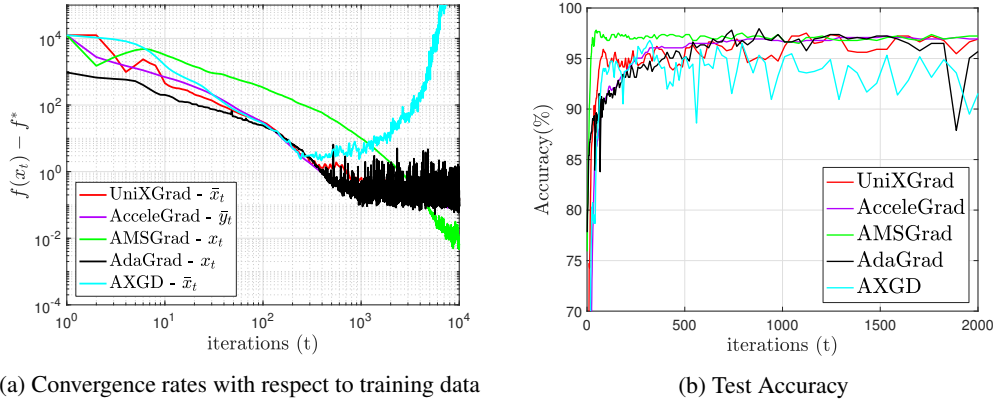

(a) Convergence rates with respect to training data            (b) Test Accuracy

Figure 3: Convergence behavior with respect to training data and resulting test accuracies for binary classification task on breast-cancer dataset from LIBSVM Chang and Lin [2011]

## 5 Discussion and Future Work

In this paper we presented an adaptive and universal framework that achieves the optimal convergence rates in constrained convex optimization setting. To our knowledge, this is the first method that

achieves $\mathcal{O}\left(GD/\sqrt{T}\right)$ and $\mathcal{O}\left(D^2L/T^2 + \sigma D/\sqrt{T}\right)$ rates in the constrained setting, without log dependencies. Without any prior information, our algorithm adapts to smoothness of the objective function as well as the variance of the possibly noisy gradients.

One would interpret that our guarantees are extensions of [Levy et al., 2018] to the constrained setting, through a completely different algorithm and a simpler, classical analysis. Our study of their algorithm and proof strategies concludes that:

- It does not seem possible to remove $\log T$ dependency in non-smooth setting for their algorithm, due to their Lemma A.3
- Extending their algorithm to constrained setting (via projecting $y$ sequence) is not trivial, as the analysis requires $y$ sequence to be unbounded (refer to their Appendix A, Eq. (16)).

As a follow up to our work, we would like to investigate three main extensions:

- Proximal version of our algorithm that could handle composite problems with nonsmooth terms, including indicator functions, in a unified manner. It seems like a rather simple extension as the main difference would be replacing optimality condition for constrained updates with that of proximal operator.
- Extending scalar adaptive learning rate to per-coordinate matrix-like preconditioner. This direction of research would help us create a robust algorithm that is applicable to non-convex problems, such as training deep neural networks.
- Adaptation to strong convexity along with smoothness and noise variance, simultaneously. A first step towards tackling this open problem is proving an improved rate of $O(1/T^2 + \sigma/T)$ for smooth and strongly convex problems, with stochastic gradients.

## Acknowledgment

AK and VC are supported by the European Research Council (ERC) under the European Union's Horizon 2020 research and innovation programme (grant agreement no 725594 - time-data) and the Swiss National Science Foundation (SNSF) under grant number 200021_178865 / 1.

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
