[Supplementary Material]

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

# A  Proof of regret-to-rate conversion

First, we discuss a generic scheme that enables us to relate our weighted regret bounds to optimality gap, hence the convergence rate. Once again, note that our analysis borrows tools and techniques from online learning literature and applies them to offline optimization setup. In essence, our conversion scheme applies to a special setting, where the convex loss is fixed across iterations. Let us give the respective Lemma and its proof.

**Lemma 1.** *Consider weighted average $\bar{x}_t$ as in Eq. (6). Let $R_T(x_*) = \sum_{t=1}^{T} \alpha_t \langle x_t - x_*, g_t \rangle$ denote the weighted regret after T iterations, $\alpha_t = t$ and $g_t = \nabla f(\bar{x}_t)$. Then,*

$$f(\bar{x}_T) - f(x_*) \leq \frac{2R_T(x_*)}{T^2}.$$

*Proof.* Let's define $A_t = \sum_{i=1}^{t} \alpha_i$. Then, by definition, we could express $x_t$ as

$$x_t = \frac{A_t}{\alpha_t} \bar{x}_t - \frac{A_{t-1}}{\alpha_t} \bar{x}_{t-1}. \tag{14}$$

Then, use Eq. (14) and replace $g_t$ by $\nabla f(\bar{x}_t)$ in the weighted regret expression, i.e.

$$
\begin{aligned}
\sum_{t=1}^{T} \alpha_t \langle x_t - x_*, \nabla f(\bar{x}_t) \rangle &= \sum_{t=1}^{T} \alpha_t \left\langle \frac{A_t}{\alpha_t} \bar{x}_t - \frac{A_{t-1}}{\alpha_t} \bar{x}_{t-1} - x_*, \nabla f(\bar{x}_t) \right\rangle \\
&= \sum_{t=1}^{T} \alpha_t \left\langle \frac{A_t}{\alpha_t} (\bar{x}_t - x_*) - \frac{A_{t-1}}{\alpha_t} (\bar{x}_{t-1} - x_*), \nabla f(\bar{x}_t) \right\rangle \\
&= \sum_{t=1}^{T} A_t \langle \bar{x}_t - x_*, \nabla f(\bar{x}_t) \rangle - A_{t-1} \langle \bar{x}_{t-1} - x_*, \nabla f(\bar{x}_t) \rangle \\
&= \sum_{t=1}^{T} \left( \sum_{i=1}^{t} \alpha_i \langle \bar{x}_t - x_*, \nabla f(\bar{x}_t) \rangle \right) - \left( \sum_{i=1}^{t-1} \alpha_i \langle \bar{x}_{t-1} - x_*, \nabla f(\bar{x}_t) \rangle \right) \\
&= \sum_{t=1}^{T} \alpha_t \langle \bar{x}_t - x_*, \nabla f(\bar{x}_t) \rangle + \sum_{t=1}^{T} \sum_{i=1}^{t-1} \alpha_i \langle \bar{x}_t - \bar{x}_{t-1}, \nabla f(\bar{x}_t) \rangle \\
&\geq \sum_{t=1}^{T} \alpha_t (f(\bar{x}_t) - f(x_*)) + \sum_{t=1}^{T} \sum_{i=1}^{t-1} \alpha_i (f(\bar{x}_t) - f(\bar{x}_{t-1})),
\end{aligned}
$$

where we used gradient inequality in the last line. We also take $\alpha_0 = 0$ and $A_0 = 0$. Then, we telescope the double summation and reorganize the terms

$$
\begin{aligned}
&= \sum_{t=1}^{T} \alpha_t (f(\bar{x}_t) - f(x_*)) + \sum_{t=1}^{T-1} \alpha_t (f(\bar{x}_T) - f(\bar{x}_t)) \\
&= \alpha_T (f(\bar{x}_T) - f(x_*)) + \sum_{t=1}^{T-1} \alpha_t (f(\bar{x}_t) - f(x_*) + f(\bar{x}_T) - f(\bar{x}_t)) \\
&= \sum_{t=1}^{T} \alpha_t (f(\bar{x}_T) - f(x_*)).
\end{aligned}
$$

Having simplified the expression, we divide both sides by $A_T$ and conclude the proof. Observe that $A_T \geq \frac{T^2}{2}$, hence,

$$\sum_{t=1}^{T} \alpha_t \left( f(\bar{x}_T) - f(x_*) \right) \leq \sum_{t=1}^{T} \alpha_t \left\langle x_t - x_*, \nabla f(\bar{x}_t) \right\rangle$$

$$\frac{1}{A_T} \sum_{t=1}^{T} \alpha_t \left( f(\bar{x}_T) - f(x_*) \right) \leq \frac{1}{A_T} \sum_{t=1}^{T} \alpha_t \left\langle x_t - x_*, \nabla f(\bar{x}_t) \right\rangle$$

$$f(\bar{x}_T) - f(x_*) \leq \frac{2R_T(x_*)}{T^2}.$$

$\square$

# B  Proofs for the non-smooth setting

As we have mentioned previously, for the weighted regret analysis in the non-smooth case, i.e., $f$ is only $G$-Lipschitz, please observe that we do not exploit the precise definitions of $g_t$ and $M_t$. As far as the regret analysis is concerned, their dual norm should be bounded. However, we especially rely on the fact that $g_t = \nabla f(\bar{x}_t)$ since it is necessary to obtain converge rates from regret-like bounds using Lemma 1.

Let us bring up the following relation which we will require for the regret analysis of both smooth and non-smooth objective.

**Lemma 2.** *Let* $\{a_i\}_{i=1,\ldots,n}$ *be a sequence of non negative numbers. Then, it holds that*

$$\sqrt{\sum_{i=1}^{n} a_i} \leq \sum_{i=1}^{n} \frac{a_i}{\sqrt{\sum_{j=1}^{i} a_j}} \leq 2\sqrt{\sum_{i=1}^{n} a_i}.$$

Please refer to [Levy et al., 2018, McMahan and Streeter, 2010] for the proof of Lemma 2, which is due to induction. We will also make use of the following bound (due to Young's Inequality)

$$\alpha_t \left\| g_t - M_t \right\|_* \left\| x_t - y_t \right\| = \inf_{\rho > 0} \left\{ \frac{\rho}{2} \left\| g_t - M_t \right\|_*^2 + \frac{\alpha_t^2}{2\rho} \left\| x_t - y_t \right\|^2 \right\}. \tag{15}$$

## B.1  Proof of Theorem 1

**Theorem 1.** *Consider the constrained optimization setting in Problem (4), where* $f : \mathcal{K} \to \mathbb{R}$ *is a proper, convex and* $G$-Lipschitz function defined over compact, convex set $\mathcal{K}$. Let $x^* \in \min_{x \in \mathcal{K}} f(x)$. *Then, Algorithm 2 guarantees*

$$f(\bar{x}_T) - \min_{x \in \mathcal{K}} f(x) \leq \frac{7D\sqrt{1 + \sum_{t=1}^{T} \alpha_t^2 \left\| g_t - M_t \right\|_*^2} - D}{T^2} \leq \frac{6D}{T^2} + \frac{14GD}{\sqrt{T}}. \tag{16}$$

*Proof.*

$$\sum_{t=1}^{T} \alpha_t \left\langle x_t - x_*, g_t \right\rangle = \sum_{t=1}^{T} \underbrace{\alpha_t \left\langle x_t - y_t, g_t - M_t \right\rangle}_{(A)} + \underbrace{\alpha_t \left\langle x_t - y_t, M_t \right\rangle}_{(B)} + \underbrace{\alpha_t \left\langle y_t - x^*, g_t \right\rangle}_{(C)}.$$

**Bounding (A)**

$$\sum_{t=1}^{T} \alpha_t \left\langle x_t - y_t, g_t - M_t \right\rangle \leq \sum_{t=1}^{T} \alpha_t \left\| g_t - M_t \right\|_* \left\| x_t - y_t \right\| \quad \text{(Hölder's Inequality)}$$

$$\leq \sum_{t=1}^{T} \frac{\rho}{2} \left\| g_t - M_t \right\|_*^2 + \frac{\alpha_t^2}{2\rho} \left\| x_t - y_t \right\|^2 \quad \text{(Equation (15))}.$$

By setting $\rho = \alpha_t^2 \eta_{t+1}$, we get the following upper bound for term (A),

$$\sum_{t=1}^{T} \alpha_t \langle x_t - y_t, g_t - M_t \rangle \leq \sum_{t=1}^{T} \frac{\alpha_t^2 \eta_{t+1}}{2} \|g_t - M_t\|_*^2 + \frac{1}{2\eta_{t+1}} \|x_t - y_t\|^2$$

**Bounding (B)**

$$\sum_{t=1}^{T} \alpha_t \langle x_t - y_t, M_t \rangle \leq \sum_{t=1}^{T} \frac{1}{\eta_t} \nabla_x D_{\mathcal{R}}(x_t, y_{t-1})^T (y_t - x_t) \quad \text{(Optimality for } x_t\text{)}$$

$$= \sum_{t=1}^{T} \frac{1}{\eta_t} \left( D_{\mathcal{R}}(y_t, y_{t-1}) - D_{\mathcal{R}}(x_t, y_{t-1}) - D_{\mathcal{R}}(y_t, x_t) \right).$$

**Bounding (C)**

$$\sum_{t=1}^{T} \alpha_t \langle y_t - x^*, g_t \rangle \leq \sum_{t=1}^{T} \frac{1}{\eta_t} \nabla_x D_{\mathcal{R}}(y_t, y_{t-1})^T (x^* - y_t) \quad \text{(Optimality for } y_t\text{)}$$

$$= \sum_{t=1}^{T} \frac{1}{\eta_t} \left( D_{\mathcal{R}}(x^*, y_{t-1}) - D_{\mathcal{R}}(y_t, y_{t-1}) - D_{\mathcal{R}}(x^*, y_t) \right).$$

**Final Bound**

$$\sum_{t=1}^{T} \alpha_t \langle x_t - x_*, g_t \rangle \leq \sum_{t=1}^{T} \frac{\alpha_t^2 \eta_{t+1}}{2} \|g_t - M_t\|_*^2 + \frac{1}{2\eta_{t+1}} \|x_t - y_t\|^2$$

$$+ \frac{1}{\eta_t} \left( D_{\mathcal{R}}(x^*, y_{t-1}) - D_{\mathcal{R}}(x^*, y_t) - D_{\mathcal{R}}(x_t, y_{t-1}) - D_{\mathcal{R}}(y_t, x_t) \right)$$

$$\leq \sum_{t=1}^{T} \frac{\alpha_t^2 \eta_{t+1}}{2} \|g_t - M_t\|_*^2 + \frac{1}{2\eta_{t+1}} \|x_t - y_t\|^2$$

$$+ \frac{1}{\eta_t} \left( D_{\mathcal{R}}(x^*, y_{t-1}) - D_{\mathcal{R}}(x^*, y_t) - \frac{1}{2} \left( \|x_t - y_t\|^2 + \|x_t - y_{t-1}\|^2 \right) \right)$$

$$\leq \sum_{t=1}^{T} \frac{\alpha_t^2 \eta_{t+1}}{2} \|g_t - M_t\|_*^2 + \sum_{t=1}^{T-1} \left( \frac{1}{\eta_{t+1}} - \frac{1}{\eta_t} \right) D_{\mathcal{R}}(x^*, y_t)$$

$$+ \sum_{t=1}^{T} \left( \frac{1}{\eta_{t+1}} - \frac{1}{\eta_t} \right) \|x_t - y_t\|^2 + \frac{1}{\eta_1} D^2$$

$$\leq \sum_{t=1}^{T} \frac{\alpha_t^2 \eta_{t+1}}{2} \|g_t - M_t\|_*^2 + \sum_{t=1}^{T} \left( \frac{1}{\eta_{t+1}} - \frac{1}{\eta_t} \right) \|x_t - y_t\|^2 + \frac{D^2}{\eta_T} + \frac{D}{2}$$

$$\leq \sum_{t=1}^{T} \frac{\alpha_t^2 \eta_{t+1}}{2} \|g_t - M_t\|_*^2 + D^2 \left( \frac{2}{\eta_{T+1}} + \frac{1}{\eta_T} \right) + \frac{D}{2}$$

$$\leq D \sum_{t=1}^{T} \frac{\alpha_t^2 \|g_t - M_t\|_*^2}{\sqrt{1 + \sum_{i=1}^{t} \alpha_t^2 \|g_t - M_t\|_*^2}} + \frac{3}{2} D \sqrt{1 + \sum_{t=1}^{T} \alpha_t^2 \|g_t - M_t\|_*^2} + \frac{D}{2}$$

$$\leq \frac{7}{2} D \sqrt{1 + \sum_{t=1}^{T} \alpha_t^2 \|g_t - M_t\|_*^2} - \frac{D}{2}$$

$$\leq 3D + 7GD \sqrt{\sum_{t=1}^{T} \alpha_t^2}$$

$$\leq 3D + 7GDT^{3/2}.$$

We obtain the rate by applying Lemma 1 to the weighted regret bound above.

□

### B.2 Proof of Theorem 2

**Theorem 2.** *Consider the optimization setting in Problem (4), where $f$ is non-smooth, convex and $G$-Lipschitz. Let $\{x_t\}_{t=1,..,T}$ be a sequence generated by Algorithm 2 such that $g_t = \tilde{\nabla} f(\bar{x}_t)$ and $M_t = \tilde{\nabla} f(\tilde{z}_t)$. With $\alpha_t = t$ and learning rate as in (7), it holds that*

$$\mathbb{E}\left[f(\bar{x}_T)\right] - \min_{x \in \mathcal{K}} f(x) \leq \frac{6D}{T^2} + \frac{14GD}{\sqrt{T}}.$$

*Proof.* Similar to $\nabla f(x) \leftrightarrow \tilde{\nabla} f(x)$ notation, $\tilde{g}_t$ denotes a stochastic but unbiased estimate of $g_t$ for any $t \in [0, .., T]$. Also note that $x^* \in \min_{x \in \mathcal{K}} f(x)$. We start with weighted regret bound,

$$R_T(x_*) = \sum_{t=1}^{T} \alpha_t \langle x_t - x^*, g_t \rangle.$$

We separate $g_t$ as $\tilde{g}_t + (g_t - \tilde{g}_t)$ and re-write the above term as

$$\sum_{t=1}^{T} \alpha_t \langle x_t - x^*, g_t \rangle = \underbrace{\sum_{t=1}^{T} \alpha_t \langle x_t - x^*, \tilde{g}_t \rangle}_{(A)} + \underbrace{\sum_{t=1}^{T} \alpha_t \langle x_t - x^*, g_t - \tilde{g}_t \rangle}_{(B)}.$$

Due to unbiasedness of the gradient estimates, expected value of $\alpha_t \langle x_t - x^*, g_t - \tilde{g}_t \rangle$, conditioned on the average iterate $\bar{x}_t$ evaluates to 0. We will only need to bound the first summation whose analysis is identical to its deterministic counterpart up to replacing $g_t$ with $\tilde{g}_t$, and $M_t$ with $\tilde{M}_t$. Hence, term (A) is upper bounded by $6D + 14GDT^{3/2}$.

In addition to the setup in the deterministic setting, we put forth the assumption that stochastic gradients have bounded norms, which is natural in the constrained optimization framework. Using Lemma 1, we translate the regret bound into the convergence rate, i.e,

$$\mathbb{E}\left[f(\bar{x}_T)\right] - \min_{x \in \mathcal{K}} f(x) \leq \frac{6D}{T^2} + \frac{14GD}{\sqrt{T}}.$$

□

## C   Proofs for the smooth setting

We will now introduce an additional assumption that $f$ is $L$-smooth (see Eq. (2)). In this section, we provide the weighted regret analysis for smooth functions in the presence of deterministic and stochastic oracles and convert these bound into suboptimality gap via our regret-to-rate scheme.

### C.1   Proof of Theorem 3

**Theorem 3.** *Consider the constrained optimization setting in Problem (4), where $f : \mathcal{K} \to \mathbb{R}$ is a proper, convex and $L$-smooth function defined over compact, convex set $\mathcal{K}$. Let $x^* \in \min_{x \in \mathcal{K}} f(x)$. Then, Algorithm 2 ensures the following*

$$f(\bar{x}_T) - \min_{x \in \mathcal{K}} f(x) \leq \frac{20\sqrt{7}D^2 L}{T^2}. \tag{17}$$

*Proof.* Recall the regret analysis for the non-smooth, convex objective

$$R_T(x_*) \leq \frac{1}{2} \sum_{t=1}^{T} \eta_{t+1} \alpha_t^2 \|g_t - M_t\|_*^2 + \frac{1}{\eta_{t+1}} \|x_t - y_t\|^2$$

$$+ \sum_{t=1}^{T} \frac{1}{\eta_t} \left( D_{\mathcal{R}}(x^*, y_{t-1}) - D_{\mathcal{R}}(x^*, y_t) - \frac{1}{2} \left( \|x_t - y_t\|^2 + \|x_t - y_{t-1}\|^2 \right) \right)$$

$$\leq \frac{1}{2} \sum_{t=1}^{T} \eta_{t+1} \alpha_t^2 \|g_t - M_t\|_*^2 + \frac{1}{2} \sum_{t=1}^{T} \left( \frac{1}{\eta_{t+1}} - \frac{1}{\eta_t} \right) \|x_t - y_t\|^2 + \sum_{t=1}^{T-1} \left( \frac{1}{\eta_{t+1}} - \frac{1}{\eta_t} \right) D_{\mathcal{R}}(x^*, y_t)$$

$$- \frac{1}{2} \sum_{t=1}^{T} \frac{1}{\eta_t} \|x_t - y_{t-1}\|^2 + \frac{D^2}{\eta_1}$$

$$= \frac{1}{2} \sum_{t=1}^{T} \eta_{t+1} \alpha_t^2 \|g_t - M_t\|_*^2 + \frac{1}{2} \sum_{t=1}^{T} \left( \frac{1}{\eta_{t+1}} - \frac{1}{\eta_t} \right) \|x_t - y_t\|^2 + \frac{1}{2} \sum_{t=1}^{T} \left( \frac{1}{\eta_{t+1}} - \frac{1}{\eta_t} \right) \|x_t - y_{t-1}\|^2$$

$$+ \sum_{t=1}^{T-1} \left( \frac{1}{\eta_{t+1}} - \frac{1}{\eta_t} \right) D_{\mathcal{R}}(x^*, y_t) - \frac{1}{2} \sum_{t=1}^{T} \frac{1}{\eta_{t+1}} \|x_t - y_{t-1}\|^2 + \frac{D^2}{\eta_1}$$

$$\leq \frac{1}{2} \sum_{t=1}^{T} \eta_{t+1} \alpha_t^2 \|g_t - M_t\|_*^2 - \frac{1}{2} \sum_{t=1}^{T} \frac{1}{\eta_{t+1}} \|x_t - y_{t-1}\|^2 + D^2 \left( \frac{2}{\eta_{T+1}} + \frac{1}{\eta_T} + \frac{1}{\eta_1} \right).$$

The key challenge in this analysis is to exploit the negative term, i.e., $-\frac{1}{2} \sum_{t=1}^{T} \frac{1}{\eta_{t+1}} \|x_t - y_{t-1}\|^2$, such that we could tighten the regret bound from non-smooth analysis. Using the smoothness of $f$ and that $\alpha_t = t$, $A_t = \sum_{i=1}^{t} \alpha_t$, $g_t = \nabla f(\bar{x}_t)$ and $M_t = \nabla f(\tilde{z}_t)$

$$\|g_t - M_t\|_*^2 \leq \frac{L^2 \alpha_t^2}{A_t^2} \|x_t - y_{t-1}\|^2$$

$$= \frac{4L^2 t^2}{t^2(t+1)^2} \|x_t - y_{t-1}\|^2$$

$$= \frac{4L^2}{\alpha_{t+1}^2} \|x_t - y_{t-1}\|^2$$

$$\leq \frac{4L^2}{\alpha_t^2} \|x_t - y_{t-1}\|^2.$$

Hence,

$$-\frac{1}{\eta_{t+1}} \|x_t - y_{t-1}\|^2 \leq -\frac{\alpha_t^2}{4L^2 \eta_{t+1}} \|g_t - M_t\|_*^2.$$

After applying this upper bound and regrouping the terms we have

$$R_T(x_*) \leq \frac{1}{2} \sum_{t=1}^{T} \left( \eta_{t+1} - \frac{1}{4L^2 \eta_{t+1}} \right) \alpha_t^2 \|g_t - M_t\|_*^2 + D^2 \left( \frac{2}{\eta_{T+1}} + \frac{1}{\eta_T} + \frac{1}{\eta_1} \right).$$

Define that $\tau^* = \max \left\{ t \in \{1, ..., T\} : \frac{1}{\eta_{t+1}^2} \leq 7L^2 \right\}$ such that $\forall t > \tau^*, \eta_{t+1} - \frac{1}{4L^2 \eta_{t+1}} \leq -\frac{3}{4} \eta_{t+1}$. We can rewrite the above term as

$$R_T(x_*) \leq \frac{1}{2}\left(\sum_{t=1}^{\tau^*}\left(\eta_{t+1}-\frac{1}{4L^2\eta_{t+1}}\right)\alpha_t^2\|g_t-M_t\|_*^2 + \sum_{t=\tau^*+1}^{T}\left(\eta_{t+1}-\frac{1}{4L^2\eta_{t+1}}\right)\alpha_t^2\|g_t-M_t\|_*^2\right)$$

$$+\frac{3D^2}{\eta_{T+1}}+\frac{D^2}{\eta_1}$$

$$\leq \underbrace{\frac{1}{2}\sum_{t=1}^{\tau^*}\eta_{t+1}\alpha_t^2\|g_t-M_t\|_*^2+\frac{D}{2}}_{(A)} + \underbrace{\frac{3D^2}{\eta_{T+1}}-\frac{3}{4}\sum_{t=\tau^*+1}^{T}\eta_{t+1}\alpha_t^2\|g_t-M_t\|_*^2}_{(B)}.$$

**Bounding (A):**   We will simply need to use the definition of $\tau^*$ and Lemma 2

$$\frac{1}{2}\sum_{t=1}^{\tau^*}\eta_{t+1}\alpha_t^2\|g_t-M_t\|_*^2+\frac{D}{2} = D\sum_{t=1}^{\tau^*}\frac{\alpha_t^2\|g_t-M_t\|_*^2}{\sqrt{1+\sum_{i=1}^{t}\alpha_i^2\|g_i-M_i\|_*^2}}+\frac{D}{2}$$

$$\leq 2D\sqrt{1+\sum_{t=1}^{\tau^*}\alpha_t^2\|g_t-M_t\|_*^2}$$

$$=\frac{4D^2}{\eta_{\tau^*+1}}$$

$$\leq 4\sqrt{7}D^2L.$$

**Bounding (B):**

$$(B) \leq \frac{3D}{2}\left(\sqrt{1+\sum_{t=1}^{T}\alpha_t^2\|g_t-M_t\|_*^2} - \sum_{t=\tau^*+1}^{T}\frac{\alpha_t^2\|g_t-M_t\|_*^2}{\sqrt{1+\sum_{i=1}^{t}\alpha_i^2\|g_i-M_i\|_*^2}}\right)$$

$$\leq \frac{3D}{2}+\frac{3D}{2}\left(\sum_{t=1}^{T}\frac{\alpha_t^2\|g_t-M_t\|_*^2}{\sqrt{1+\sum_{i=1}^{t}\alpha_i^2\|g_i-M_i\|_*^2}} - \sum_{t=\tau^*+1}^{T}\frac{\alpha_t^2\|g_t-M_t\|_*^2}{\sqrt{1+\sum_{i=1}^{t}\alpha_i^2\|g_i-M_i\|_*^2}}\right)$$

$$\leq \frac{3D}{2}+\frac{3D}{2}\sum_{t=1}^{\tau^*}\frac{\alpha_t^2\|g_t-M_t\|_*^2}{\sqrt{1+\sum_{i=1}^{t}\alpha_i^2\|g_i-M_i\|_*^2}}$$

$$\leq 3D\sqrt{1+\sum_{t=1}^{\tau^*}\alpha_i^2\|g_i-M_i\|_*^2}$$

$$=\frac{6D^2}{\eta_{\tau^*+1}}$$

$$\leq 6\sqrt{7}D^2L.$$

**Final Bound:**   What remains is to simply bring the term (A) and (B) together.

$$R_T(x_*) \leq \frac{1}{2}\sum_{t=1}^{\tau^*}\eta_{t+1}\alpha_t^2\|g_t-M_t\|_*^2+\frac{D}{2}+\frac{3D^2}{\eta_{T+1}}-\frac{3}{4}\sum_{t=\tau^*+1}^{T}\eta_{t+1}\alpha_t^2\|g_t-M_t\|_*^2$$

$$\leq 10\sqrt{7}D^2L.$$

We conclude the proof by applying Lemma 1 and get $f(\bar{x}_T)-\min_{x\in\mathcal{K}}f(x)\leq\frac{20\sqrt{7}D^2L}{T^2}$.

$\square$

## C.2 Proof of Theorem 4

In this setting, we will make an additional, but classical, bounded variance assumption on the stochastic gradient oracles. Recall the bounded variance assumption in Eq. (10) and let us define $\xi_t = (\tilde{g}_t - \tilde{M}_t) - (g_t - M_t)$. Since $\|\xi_t\|_*^2 \leq 2\|\tilde{g}_t - g_t\|_*^2 + 2\|\tilde{M}_t - M_t\|_*^2$, we can write,

$$\mathbb{E}\left[\|\xi_t\|_*^2 | \bar{x}_t\right] \leq 4\sigma^2. \tag{18}$$

Next, we will present our final convergence theorem.

**Theorem 4.** *Consider the optimization setting in Problem (4), where $f$ is L-smooth and convex. Let $\{x_t\}_{t=1,...,T}$ be a sequence generated by Algorithm 2 such that $g_t = \tilde{\nabla} f(\bar{x}_t)$ and $M_t = \tilde{\nabla} f(\tilde{z}_t)$. With $\alpha_t = t$ and learning rate as in (7), it holds that*

$$\mathbb{E}\left[f(\bar{x}_T)\right] - \min_{x \in \mathcal{K}} f(x) \leq \frac{224\sqrt{14}D^2 L}{T^2} + \frac{14\sqrt{2}\sigma D}{\sqrt{T}}.$$

*Proof.* We start out with weighted regret, the same way as in Theorem 2

$$\sum_{t=1}^{T} \alpha_t \langle x_t - x^*, g_t \rangle \leq \underbrace{\sum_{t=1}^{T} \alpha_t \langle x_t - x^*, \tilde{g}_t \rangle}_{(A)} + \underbrace{\sum_{t=1}^{T} \alpha_t \langle x_t - x^*, g_t - \tilde{g}_t \rangle}_{(B)}.$$

We already know that term (B) is zero in expectation. Following the proof steps of Theorem 2, we could upper bound term (A) as

$$\leq \frac{1}{2} \sum_{t=1}^{T} \eta_{t+1} \alpha_t^2 \|\tilde{g}_t - \tilde{M}_t\|_*^2 - \frac{1}{2} \sum_{t=1}^{T} \frac{1}{\eta_{t+1}} \|x_t - y_{t-1}\|^2 + D^2 \left(\frac{3}{\eta_{T+1}} + \frac{1}{\eta_1}\right)$$

$$= \frac{D}{2} + D \sum_{t=1}^{T} \frac{\alpha_t^2 \|\tilde{g}_t - \tilde{M}_t\|_*^2}{\sqrt{1 + \sum_{i=1}^{t} \alpha_i^2 \|\tilde{g}_t - \tilde{M}_t\|_*^2}} + \frac{3D}{2} \sqrt{1 + \sum_{t=1}^{T} \alpha_t^2 \|\tilde{g}_t - \tilde{M}_t\|_*^2} - \sum_{t=1}^{T} \frac{\|x_t - y_{t-1}\|^2}{2\eta_{t+1}}$$

$$\leq \frac{7D}{2} \sqrt{1 + \sum_{t=1}^{T} \alpha_t^2 \|\tilde{g}_t - \tilde{M}_t\|_*^2} - \frac{1}{2} \sum_{t=1}^{T} \frac{1}{\eta_{t+1}} \|x_t - y_{t-1}\|^2.$$

Now lets denote,

$$B_t^2 := \min\{\|g_t - M_t\|_*^2, \|\tilde{g}_t - \tilde{M}_t\|_*^2\},$$

as well as an auxiliary learning rate which we will only use for the analysis

$$\tilde{\eta}_t = \frac{2D}{\sqrt{1 + \sum_{i=1}^{t-1} \alpha_i^2 B_i^2}}. \tag{19}$$

Clearly, for any $t \in [T]$ we have $1/\tilde{\eta}_t \leq 1/\eta_t$, and therefore,

$$-\frac{1}{\eta_{t+1}} \|g_t - M_t\|_*^2 \leq -\frac{1}{\tilde{\eta}_{t+1}} B_t^2. \tag{20}$$

Also, for $\xi_t = (\tilde{g}_t - \tilde{M}_t) - (g_t - M_t)$, we can write,

$$\|\tilde{g}_t - \tilde{M}_t\|_*^2 \leq 2\|g_t - M_t\|_*^2 + 2\|\xi_t\|_*^2. \tag{21}$$

Thus,

$$
\begin{aligned}
\|\tilde{g}_t - \tilde{M}_t\|_*^2 &= B_t^2 + \left( \|\tilde{g}_t - \tilde{M}_t\|_*^2 - \min\{\|g_t - M_t\|_*^2, \|\tilde{g}_t - \tilde{M}_t\|_*^2\} \right) \\
&= B_t^2 + \max\{0, \|\tilde{g}_t - \tilde{M}_t\|_*^2 - \|g_t - M_t\|_*^2\} \\
&\leq B_t^2 + B_t^2 + 2\|\xi_t\|_*^2 \\
&= 2B_t^2 + 2\|\xi_t\|_*^2,
\end{aligned}
$$

where the last inequality is due to the fact that if $\|\tilde{g}_t - \tilde{M}_t\|_*^2 \geq \|g_t - M_t\|_*^2$, then $B_t^2 := \|g_t - M_t\|_*^2$. Then, we combine this with Eq. (21) to deduce that $\|\tilde{g}_t - \tilde{M}_t\|_*^2 - \|g_t - M_t\|_*^2 \leq B_t^2 + 2\|\xi_t\|_*^2$.

We will take conditional expectation after we simplify the expression. Now, we plug Eq. (20) and (21) into above bound,

$$
\begin{aligned}
&\leq \frac{7D}{2} \sqrt{1 + 2\sum_{t=1}^{T} \alpha_t^2 B_t^2 + \alpha_t^2 \|\xi_t\|_*^2} - \frac{1}{2} \sum_{t=1}^{T} \frac{1}{4L^2 \tilde{\eta}_{t+1}} \alpha_t^2 B_t^2 \\
&\leq \frac{7D}{\sqrt{2}} \sqrt{\sum_{t=1}^{T} \alpha_t^2 \|\xi_t\|_*^2} + \frac{7D}{2} \sqrt{1 + 2\sum_{t=1}^{T} \alpha_t^2 B_t^2} - \frac{1}{2} \sum_{t=1}^{T} \frac{1}{4L^2 \tilde{\eta}_{t+1}} \alpha_t^2 B_t^2 \\
&\leq \frac{7D}{2} + \frac{7D}{\sqrt{2}} \sqrt{\sum_{t=1}^{T} \alpha_t^2 \|\xi_t\|_*^2} + 7D \sum_{t=1}^{T} \frac{\alpha_t^2 B_t^2}{\sqrt{1 + 2\sum_{i=1}^{t} \alpha_i^2 B_i^2}} - \frac{1}{2} \sum_{t=1}^{T} \frac{1}{4L^2 \tilde{\eta}_{t+1}} \alpha_t^2 B_t^2 \\
&\leq \frac{7D}{2} + \frac{7D}{\sqrt{2}} \sqrt{\sum_{t=1}^{T} \alpha_t^2 \|\xi_t\|_*^2} + 7D \sum_{t=1}^{T} \frac{\alpha_t^2 B_t^2}{\sqrt{1 + \sum_{i=1}^{t} \alpha_i^2 B_i^2}} - \frac{1}{2} \sum_{t=1}^{T} \frac{1}{4L^2 \tilde{\eta}_{t+1}} \alpha_t^2 B_t^2 \\
&\leq \underbrace{\frac{7}{2} \sum_{t=1}^{T} \left( \tilde{\eta}_{t+1} - \frac{1}{28L^2 \tilde{\eta}_{t+1}} \right) \alpha_t^2 B_t^2 + \frac{7D}{2}}_{(A)} + \underbrace{\frac{7D}{\sqrt{2}} \sqrt{\sum_{t=1}^{T} \alpha_t^2 \|\xi_t\|_*^2}}_{(B)}.
\end{aligned}
$$

**Bounding (A):** We will make use of the exact same approach as we did in Theorem 3, where we defined an auxiliary time variable $\tau^*$ to characterize the behavior of the learning rate.

Now, let us denote $\tau^* = \max \left\{ t \in \{1, ..., T\} \ : \ \frac{1}{\tilde{\eta}_{t+1}^2} \leq 56L^2 \right\}$. It implies that

$$
\tilde{\eta}_{t+1} - \frac{1}{28L^2 \tilde{\eta}_{t+1}} \leq -\tilde{\eta}_{t+1}, \quad \forall t > \tau^*. \tag{22}
$$

Then, we could proceed as

$$
\begin{aligned}
\text{(A)} &= \frac{7}{2} \sum_{t=1}^{\tau^*} \left( \tilde{\eta}_{t+1} - \frac{1}{28 L^2 \tilde{\eta}_{t+1}} \right) \alpha_t^2 B_t^2 + \frac{7}{2} \sum_{t=\tau^*+1}^{T} \left( \tilde{\eta}_{t+1} - \frac{1}{28 L^2 \tilde{\eta}_{t+1}} \right) \alpha_t^2 B_t^2 + \frac{7D}{2} \\
&\leq \frac{7}{2} \sum_{t=1}^{\tau^*} \tilde{\eta}_{t+1} \alpha_t^2 B_t^2 - \frac{7}{2} \sum_{t=\tau^*+1}^{T} \tilde{\eta}_{t+1} \alpha_t^2 B_t^2 + \frac{7D}{2} \\
&\leq \frac{7}{2} \sum_{t=1}^{\tau^*} \tilde{\eta}_{t+1} \alpha_t^2 B_t^2 + \frac{7D}{2} \\
&= 7D \sum_{t=1}^{\tau^*} \frac{\alpha_t^2 B_t^2}{\sqrt{1 + \sum_{i=1}^{t} \alpha_i^2 B_i^2}} + \frac{7D}{2} \\
&\leq 14D \sqrt{1 + \sum_{t=1}^{\tau^*} \alpha_t^2 B_t^2} \\
&\leq \frac{28 D^2}{\tilde{\eta}_{\tau^*+1}} \\
&\leq 112 \sqrt{14} D^2 L.
\end{aligned}
$$

**Bounding (B):**  Following bounded variance definition in Eq. (10), we can write $\mathbb{E}[\|\xi_t\|_*^2] \leq 4\sigma^2$. After taking expected value conditioned on $\bar{x}_t$, we simply use Jensen's inequality to complete the proof

$$
\begin{aligned}
\mathbb{E} \left[ \frac{7D}{\sqrt{2}} \sqrt{\sum_{t=1}^{T} \alpha_t^2 \|\xi_t\|_*^2} \;\middle|\; \bar{x}_t \right] &\leq \frac{7D}{\sqrt{2}} \sqrt{\mathbb{E} \left[ \sum_{t=1}^{T} \alpha_t^2 \|\xi_t\|_*^2 \;\middle|\; \bar{x}_t \right]} \\
&= \frac{7D}{\sqrt{2}} \sqrt{\sum_{t=1}^{T} \alpha_t^2 \mathbb{E} \left[ \|\xi_t\|_*^2 \;\middle|\; \bar{x}_t \right]} \\
&\leq \frac{7D}{\sqrt{2}} \sqrt{\sum_{t=1}^{T} 4 \alpha_t^2 \sigma^2} \\
&\leq \frac{14 D \sigma}{\sqrt{2}} \sqrt{T^3} \\
&= \frac{14 \sigma D T^{3/2}}{\sqrt{2}}.
\end{aligned}
$$

Finally, we combine all these bounds together and feed them through Lemma 1 to obtain the final rate.

$$
\mathbb{E} \left[ f(\bar{x}_T) \right] - \min_{x \in \mathcal{K}} f(x) \leq \frac{224 \sqrt{14} D^2 L}{T^2} + \frac{14 \sqrt{2} \sigma D}{\sqrt{T}}.
$$

$\square$