[Reviews · NeurIPS 2019]

Reviewer 1



originality: I don't know of any other results of this form, although the non-adaptive problem has been solved previously, and the unconstrained problem has been solved up to logs. minor nit: I don't think the version of lemma 1 in cutkosky2019 has log dependencies - it looks like lemma 1 is a special case of theorem 1 in cutkosky2019. quality: This paper presents a clean solution to an important open problem in adaptive convex optimization. The techniques seem interesting and may be useful for other analyses. I liked the use of optimism here and I think this in particular is an under-utilized type of guarantee that might have more applications. clarity: the paper is clearly written. I found even the proofs in the appendices to be fairly painless to follow! significance: This is paper solves an interesting open problem in adaptive stochastic convex optimization and so I believe its results to be significant. -------------- I have read the author response and maintain my score.

Reviewer 2



I have read the author's response and appreciate the proof provided in this work. I updated my score to 8. ================================ The paper is very cleanly written. Problem setup, proofs and convergence guarantees are easy to read. The main result is new. The theoretical significance of this result is high as it provides a clean way to achieve acceleration for constrained smooth/nonsmooth convex optimization problems. The empirical significance is limited but it's fine as experiment improvement is not the main claim of the paper.

Reviewer 3



Overall, I think the result is nice and the analysis has some new techniques which I have not seen before. The only issue is that the authors do not consider the strongly convex case, which is more challenging. In the previous work, for example, the work by Guanghui Lan, achieving the uniformly optimal rate for non-strongly convex case is easy but for strongly convex case is much harder. I wonder if the authors can discuss the main difficulty of extending their results to the strongly convex case. Another suggestion is to explicitly define alpha_t=t. I can only find alpha_t defined in Lemma 1 and have to guess that it is still defined as alpha_t=t for the rest of the theorem and their proofs. Please define alpha just like how you define eta_t. If alpha_t varies with theorems, please define them separately. ++++++++++++++ I have read the comments from other reviewers and the responses. I am satisfied with the responses and will keep my score still at 7.

[Author Response · NeurIPS 2019]

We thank all the reviewers for their detailed comments, constructive suggestions and thorough evaluation of our work.

**Reviewer 1:**   Indeed, two main contributions of our framework are the removal of log dependencies and achieving the optimal accelerated rates for constrained problems.

We agree that our Lemma 1 is a special case of Theorem 1 of Cutkosky [2019], where $\alpha_t = t$. We would like to thank the reviewer for pointing out the mistake in our sentence "*In a concurrent work, [Cutkosky, 2019] proves a similar online-to-offline conversion bound, with log dependencies.*" We will remove "*with log dependencies*" part of the sentence in the final submission.

Regarding your comment about the space diameter, it is worth investigating how we could also adapt to the initial distance to the constraint set. It is a challenge to remove the space diameter as it appears due to classical regret analysis.

**Reviewer 2:**   We have studied the convergence analysis of Levy et al. [2018]. It is much more complicated compared to our approach, and difficult to integrate with classical techniques. Extending it to constrained setting is not straight-forward, yet we cannot see a way to remove the log dependencies in their analysis either. With our analysis technique, we achieve both of these simultaneously, and provide an alternative interpretation of acceleration with a simpler proof.

We propose in the Conclusion section that we could extend our framework for composite problems with non-smooth components, as well. For composite problems, the main difference in the analysis would be the optimality conditions used in the proof of Theorem 1, due to proximal gradient steps. This appears as a rather straight-forward extension of our paper. For Levy et al. [2018], we have no such claims. In fact, it could be a challenging task to extend their analysis for composite problems as the proof techniques are complex and unique to their framework. It is rather difficult to incorporate classical analysis tools into theirs.

**Reviewer 3:**   Our primary objective in this paper is to develop an adaptive algorithm that universally achieves optimal rates for smooth/non-smooth objectives with stochastic/deterministic first-order oracles, in the constrained setting.

We actually give the definition of $\alpha_t$ in the "Gradient weighting scheme" part, and argue that it should be of order $\Theta(t)$. We will replace this with a clearer definition, i.e., $\alpha_t = t$, as you suggest.

Regarding your comment on strong convexity, Lan and Ghadimi [2012] show that with stochastic first-order oracles and strongly convex objective, their algorithm converges with a rate of $O(1/T^2 + \sigma/T)$. In the setting of deterministic oracles the optimal rate is linear, however. Therefore, obtaining a rate of $O(\exp(-T) + \sigma^2/T)$ is not straight-forward. We also believe it is not in the main scope of this paper, but an important challenge we consider to tackle as future work.

There exist prior related work in online learning about adaptation to strong convexity, such as [Hazan et al., 2008]. It is known that the regret bound for strongly-convex sequence of functions is $\log(T)$. We could design our proof technique in parallel with such prior work and try to obtain a bound like $O(1/T^2 + \log(T)\sigma^2/T)$ with stochastic first-order oracles by adapting to strong convexity. We will consider this as future work.

To the best of our knowledge, there does not exist any universal and adaptive convex optimization method that "simultaneously" adapts to smoothness, strong convexity and noise variance while achieving the optimal rates. Thus it is, indeed, an open problem that we are willing to investigate.

# References

A. Cutkosky. Anytime online-to-batch conversions, optimism, and acceleration. *the International Conference on Machine Learning (ICML)*, June 2019.

E. Hazan, A. Rakhlin, and P. L. Bartlett. Adaptive online gradient descent. In J. C. Platt, D. Koller, Y. Singer, and S. T. Roweis, editors, *Advances in Neural Information Processing Systems 20*, pages 65–72. Curran Associates, Inc., 2008.

G. Lan and S. Ghadimi. Optimal stochastic approximation algorithms for strongly convex stochastic composite optimization i: A generic algorithmic framework. *SIAM Journal on Optimization*, 22, 11 2012.

K. Y. Levy, A. Yurtsever, and V. Cevher. Online adaptive methods, universality and acceleration. In *Neural and Information Processing Systems (NeurIPS)*, December 2018.


[Meta-Review · NeurIPS 2019]

All the reviewers agreed that the paper should be accepted in NeurIPS. Please take into account the reviewers' comments in preparing the camera-ready version.